# Resolving the data ambiguity for periodic crystals

**Daniel E Widdowson**
Department of Computer Science
University of Liverpool, Liverpool, L69 3BX
`D.E.Widdowson@liverpool.ac.uk`

**Vitaliy A Kurlin**
Department of Computer Science
University of Liverpool, Liverpool, L69 3BX
`vitaliy.kurlin@gmail.com`

## Abstract

The fundamental model of all solid crystalline materials is a periodic set of atomic centers considered up to rigid motion in Euclidean space. The major obstacle to materials discovery was highly ambiguous representations of periodic crystals that didn't allow fast and reliable comparisons and led to numerous (near-) duplicates in many databases of experimental and simulated crystals. This paper exemplarily resolves the ambiguity by invariants, which are descriptors without false negatives.

The new Pointwise Distance Distributions (PDD) is a numerical matrix with a near-linear time complexity and an exactly computable metric. The strongest theoretical result is generic completeness (absence of false positives) for all finite and periodic sets of points in any dimension. The strength of PDD is shown by 200B+ pairwise comparisons of all periodic structures in the world's largest collection (Cambridge Structural Database) of existing materials over two days on a modest desktop.

## 1   Motivations for resolving the ambiguity challenge for periodic crystals

This paper resolves the long-standing challenge of ambiguous data representation for periodic structures that model all solid crystalline materials (periodic crystals). Any real crystal is best modeled as a periodic set $S \subset \mathbb{R}^n$ of points at all atomic centers, whose positions have a physical meaning and are determined via diffraction patterns. Edges between points are excluded because they only abstractly represent inter-atomic bonds depending on thresholds for distances and angles [29].

The simplest example is a *lattice* $\Lambda \subset \mathbb{R}^n$ consisting of all integer linear combinations of a basis whose vectors span a *unit cell* $U$. Fig. 1 shows omly a few of infinitely many possible choices of $U$.

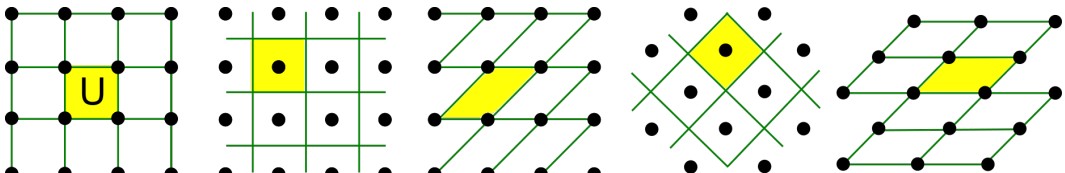

Figure 1: These isometric lattices are given by different cells and motifs. **1st**: $U = \langle (1,0), (0,1) \rangle$, $M = \{(0,0)\}$. **2nd**: $U = \langle (1,0), (0,1) \rangle$, $M = \{(\frac{1}{2}, \frac{1}{2})\}$. **3rd**: $U = \langle (1,0), (1,1) \rangle$, $M = \{(0,0)\}$. **4th**: $U = \langle (\frac{1}{\sqrt{2}}, \frac{1}{\sqrt{2}}), (-\frac{1}{\sqrt{2}}, \frac{1}{\sqrt{2}}) \rangle$, $M = \{(\frac{1}{2}, \frac{1}{2})\}$. **5th**: $U = \langle (\sqrt{2}, 0), (\frac{1}{\sqrt{2}}, \frac{1}{\sqrt{2}}) \rangle$, $M = \{(0,0)\}$.

Materials discovery still relies on trial-and-error because periodic crystals are traditionally represented by non-invariants (descriptors with false negatives) or discontinuous invariants such as symmetry groups that break down under tiny perturbations. These conventional descriptions cannot identify fraudulent structures in experimental datasets that keep depositing numerous (near-) duplicates without reliable tools for justified comparisons [30]. The ambiguity challenge will be rigorously stated in Problem 1.1 as a classification of periodic sets up to isometry preserving the crystal rigidity.

36th Conference on Neural Information Processing Systems (NeurIPS 2022).

Fig. 1 illustrates the first obstacle: the same lattice can be generated by infinitely many different bases or unit cells, so distinguishing only lattices up to isometry is already non-trivial. Any *periodic point set S* can be defined as a sum $\Lambda + M = \{\vec{v} + \vec{p} : v \in \Lambda, p \in M\}$, where a *motif M* is a finite set of points in the basis of $U$. Any lattice $\Lambda$ is considered as a periodic set with a 1-point motif $M = \{p\}$. A single point $p$ can be arbitrarily chosen in a unit cell $U$, see the first two pictures of Fig. 1. Basis vectors of $U$ and atomic coordinates of motif points (atomic centers) in $M$ constitute the data found in conventional Crystallographic Information File (CIF). Fig. 2 (left) shows the ambiguity of the CIF pair $(\Lambda, M)$ even if a basis of $U$ is fixed. The recent work by Widdowson *et al.* [67] rigorously stated the problem to continuosly classify periodic point sets up to isometry. An *isometry* of Euclidean space $\mathbb{R}^n$ is any map that maintains inter-point distances. Any orientation-preserving isometry can be realized as a continuous rigid motion, for example any composition of translations and rotations in $\mathbb{R}^3$. This equivalence is most natural for finite and periodic sets that represent real rigid structures.

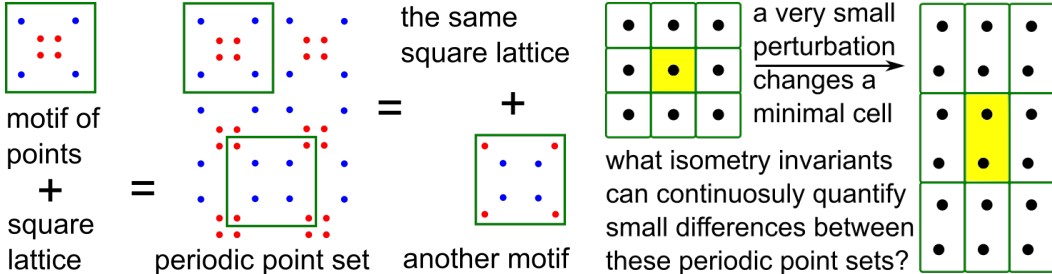

Figure 2: **Left** : even for a fixed cell of a lattice $\Lambda$, different motifs $M$ can define isometric periodic sets $\Lambda + M$. **Right**: for almost any perturbation, the symmetry group and (the minimum volume of) any reduced cell discontinuously change, which justifies continuity condition (1.1d) in Problem 1.1.

Crystals can be reliably distinguished up to isometry only by an *isometry invariant I* that takes the same value on all isometric sets. So $I$ has no *false negatives* such that $I(S) \neq I(Q)$ for different representations of isometric crystals $S \cong Q$. If a descriptor has false negatives, we can make *no reliable conclusions* because different inputs as in Fig. 1 and 2 (left) can reprsent isometric crystals.

Hence, non-invariants cannot be used to justifiably compare crystals [7]. It suffices to classify crystals up to isometry $f$ including reflections because, for any basis $v_1, \ldots, v_n$ of $\mathbb{R}^n$, $f$ preserves orientation if the determinant $\det(f(v_1), \ldots, f(v_n)) > 0$ for the images of $v_1, \ldots, v_n$ in the same basis.

The traditional approach to identifying a periodic crystal is to use its conventional or reduced cell [40, section 9.3]. This reduced cell has been known to be discontinuous under perturbations [44] even for 2D lattices when a motif $M$ is one point. More formally, [26, section 1] and [67, Theorem 15] proved that a reduced cell cannot be continuously defined for all lattices. Discontinuity of many discrete invariants such as symmetry groups becomes clearer in Fig. 2 (right) showing that even real-valued invariants struggle to continuously quantify the similarity between nearly identical sets. Under perturbations of points, a minimal cell $U$ can double in size, while the density remains constant.

**Problem 1.1.** *Find a function I on all periodic sets of unlabeled points in $\mathbb{R}^n$ such that*

*(1.1a)* invariance: *if any periodic point sets $S \cong Q$ are isometric in $\mathbb{R}^n$, then $I(S) = I(Q)$, so the invariant I has* no false negatives;

*(1.1b)* completeness: *if $I(S) = I(Q)$ for any periodic point sets $S, Q$, then $S \cong Q$ are isometric, so the invariant I has* no false positives;

*(1.1c)* metric: *a distance $d$ between values of I satisfies all axioms; 1) $d(I_1, I_2) = 0$ if and only if $I_1 = I_2$, 2) symmetry $d(I_1, I_2) = d(I_2, I_1)$, 3) triangle inequality $d(I_1, I_2) + d(I_2, I_3) \geq d(I_1, I_3)$;*

*(1.1d)* Lipschitz continuity: *if $Q$ is obtained from a periodic set $S \subset \mathbb{R}^n$ by shifting each point of $S$ by at most $\varepsilon$, then $d(I(S), I(Q)) \leq C\varepsilon$ for a constant $C$ and any periodic point sets $S, Q \subset \mathbb{R}^n$;*

*(1.1e)* computability: *the invariant $I$, the metric $d$ and verification of $I(S) = I(Q)$ should be computed in a near-linear time in the number of motif points of periodic sets for a fixed dimension $n$;*

*(1.1f)* inverse design: *any periodic point set $S \subset \mathbb{R}^n$ can be reconstructed from its invariant $I(S)$.* ∎

Problem 1.1 is the ultimate *data science challenge* for all periodic crystals $S$ whose non-invariant input (a cell basis and a motif) should be transformed into a complete invariant $I(S)$, which uniquely and unambiguously represents any $S$. Such a complete invariant can be considered as a *materials genome* [22] or a *DNA-style code* that also allows an explicit reconstruction for any periodic crystal.

For example, computer vision tries to identify humans or other objects such as road signs by using pixel-based images as input. Similar to other real objects, any periodic crystal can be given by (infinitely) many inputs. Problem 1.1 is an example of the data ambiguity challenge for almost any application that needs to uniquely identify a real object. Biology has partially solved this challenge by a DNA code, though a living creature cannot be easily created from such a code. Condition (1.1f) goes further and requires a full reconstruction of a periodic crystal $S$ from its DNA-style code $I(S)$.

The proposed solution to Problem 1.1 is the isometry invariant $I$ called the Pointwise Distance Distribution PDD. Theorems 3.2, 4.3, 5.1 , 4.4 prove that PDD satisfies all conditions of Problem 1.1, even (1.1b) at least for generic sets. More exactly, Theorem 4.4 shows that any periodic point set $S = \Lambda + M \subset \mathbb{R}^n$ in a general position can be explicitly reconstructed from PDD and its lattice $\Lambda$.

The strength of PDD was experimentally checked for all 660K+ periodic crystals in the world's largest collection CSD (Cambridge Structural Database). Despite the CSD being curated to contain only real and distinct structures [30], the new invariants identified several pairs of duplicates. All the underlying publications are now being investigated for data integrity by five journals, see section 6.

Problem 1.1 is stated in the hardest scenario when points are unordered and unlabeled because many real crystals have identical compositions. For example, diamond and graphite (whose 2-dimensional layer is graphene) both consist of pure carbon but have vastly different physical properties.

Conditions (1.1cd) for a continuous metric are stronger than a complete classification in (1.1ab): detecting an isometry gives a discontinuous metric $d(S,Q) = 1$ (or another positive number) for all non-isometric $S \not\cong Q$ even if $S, Q$ are near duplicates as in Fig. 2 (right). Continuity under perturbations is practically important because atoms vibrate, and any real measurement of a crystal produces slightly different coordinates of its unit cell and motif. Any simulation of periodic structures introduces floating point errors because of inevitable approximations by iterative optimization. Thousands of near-duplicates are routinely produced, though only a few structures are synthesized. Five real structures of 5679 predicted on a supercomputer over 12 weeks are a typical example [29]. This 'embarrassment of over-prediction' [52] wastes time and resources to run simulations and then analyze results, often by visual inspection, because of a lack of invariants that solve Problem 1.1.

Computability condition (1.1e) avoids the trivial function $I(S) = S$ in Problem 1.1. Inverse design in (1.1f) allows one to replace the traditional blind sampling (of ambiguous cells and motifs leading to (near-) duplicates via optimization) with a guided exploration of the crystal space parameterized by complete and reversible invariants. Section 2 shows that the state-of-the-art tools remain stuck with conditions (1.1ab) while the new invariants satisfy the stronger practical requirements (1.1cdef).

## 2 A review of the related state-of-the-art on comparing periodic point sets

**Computational Geometry** methods studied isometry questions related to Problem 1.1 for finite sets of points. The fastest algorithm to test an isometry between $m$-point sets in $\mathbb{R}^n$ runs in time $O(m^{\lceil n/3 \rceil} \log m)$ by [13], which can be improved to $O(m \log m)$ in $\mathbb{R}^4$ [41]. Significant results on matching bounded rigid shapes and registration of finite point sets were obtained in [56, 33, 31, 25]. These methods focus on true/false answers without continuous and easily computable metrics [58, 46].

The classical isometry invariant of a finite set in any metric space is the distribution of pairwise distances who completeness in general position was proved in 2004 [12]. The same paper described an infinite family of singular counter-examples to the full completeness, see the non-isometric 4-point sets in Fig. 3 (left). Persistent homology is a more recent isometry invariant of finite point sets, which is continuous under perturbation but turned out to be weaker than previously anticipated [49, 27, 28, 60]. Mémoli's seminal work on *distributions of distances* [45], also known as *shape distributions* [48, 9, 38, 35, 32] for bounded metric spaces is closest to the proposed Pointwise Distance Distributions (PDD) for periodic point sets. However, Problem 1.1 is not reducible to the finite sets by taking a cube or a ball of a fixed (even very large) cut-off radius within a periodic point set. Indeed, one can easily find non-isometric subsets of the same periodic set as in Fig. 4 (left).

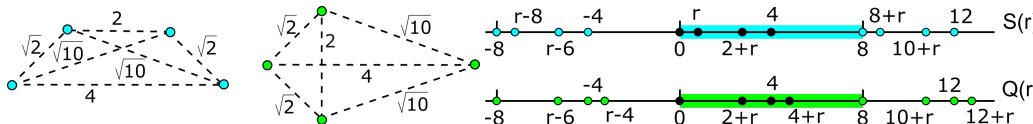

Figure 3: **Left**: point sets $K = \{(\pm 2, 0), (\pm 1, 1)\}$ and $T = \{(\pm 2, 0), (-1, \pm 1)\}$ can not be distinguished by their six pairwise distances $\sqrt{2}, \sqrt{2}, 2, \sqrt{10}, \sqrt{10}, 4$. **Right**: 1D periodic sets $S(r) = \{0, r, 2 + r, 4\} + 8\mathbb{Z}$ and $Q(r) = \{0, 2 + r, 4, 4 + r\} + 8\mathbb{Z}$ for $0 < r \le 1$ have the same Patterson function [51, p. 197, Fig. 2]. All these pairs are distinguished by PDD in section 3.

**Computer Vision** methods focused on alignment or registration of point clouds in dimensions 2, 3 [64, 53]. The latest isometry invariant [17] is defined for a scalar field $\mathbb{R}^3 \to \mathbb{R}^1$, which is more complicated than for a finite cloud. Completeness up to isometry was resolved by the invariants [42] with continuous metrics that have polymomial time in the number of points for a fixed dimension.

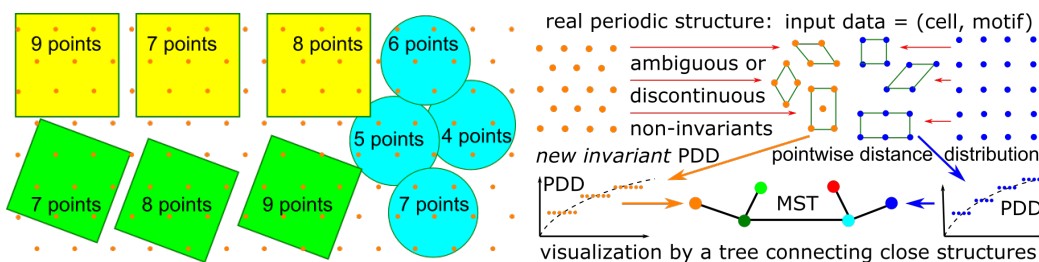

Figure 4: **Left**: isometric periodic sets have many non-isometric finite subsets that cannot be used for comparison, discontinuity under perturbations is similar to Fig. 2 (right). **Right**: an ambiguous input is transformed into PDD to visualize any dataset as a Minimum Spanning Tree (MST).

**Crystallography and Materials Science** developed many non-invariant descriptors by using quantities cell parameters and atomic coordinates in an ambiguous cell. The past approaches such as SOAP (Smooth Overlapped Atomic Positions [7]), ACSF (Atom-Centered Symmetry Functions [8]), and ACE (Atomic Cluster Expansion [24]) rely on finite subsets with a fixed number of neighbors [18] or a cut-off radius [37], see Fig. 4 (left), without continuity in 1.1(d). A reduction to finite-size subsets cannot guarantee a complete and continuous classification because, under tiny perturbations, a minimal (by volume) cell can become larger than any subset of a fixed size, see Fig. 2 (right).

Perturbations of atoms can change inter-atomic bonds depending on thresholds, which makes all graph-based descriptors [68, 20] discontinuous [6, 65] and unreliable for detecting near-duplicates. If threshold parameters change, then so do (possibly, discontinuously) the invariants, see Fig. 5.

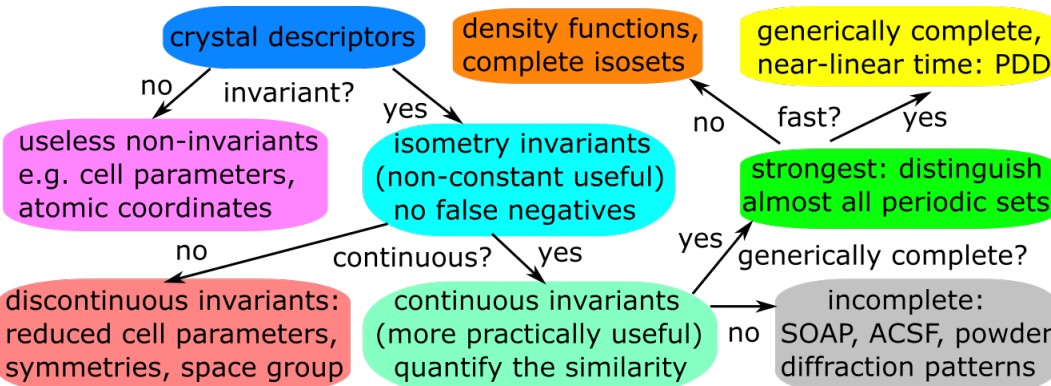

Figure 5: A zoo of crystal descriptors. A minimally useful (non-constant) invariant should have *no false negatives*. Most isometry invariants are discontinuous [70] or incomplete [50], hence fail conditions (1.1bcd). The fastest and generically complete invariants are PDD defined in this paper.

One of the oldest invariants of crystals is the X-ray diffraction pattern whose single crystal form is best for determining a 3D structure of an experimental crystal [21]. Since not all materials can be grown as single crystals, powder X-ray diffraction patterns (PXRD) are more common. All periodic structures with identical PXRDs are called *homometric* [54], see the periodic versions $S(1) = \{0, 1, 3, 4\} + 8\mathbb{Z}$ and $Q(1) = \{0, 3, 4, 5\} + 8\mathbb{Z}$ of the 4-point set $T, K$ in Fig. 3. The more general sets $S(r), Q(r)$ with the same Pair Distribution Function (PDF) will be distinguished by PDD in section 3.

The CCDC's Mercury software compares periodic structures [18] by minimizing the Root Mean Square Deviation (RMSD) of atomic positions from up to a given number $m$ (15 by default) of closest molecules in two structures. This comparison depends on many parameters (maximum number of matched molecules, thresholds for matched distances and angles), fails the triangle inequality in condition (1.1c), and is too slow for pairwise comparisons, see section 6. This RMSD tries to measure the maximum displacement of atoms only by using finite subsets, see the difficulties in Fig. 4 (left).

Any point $p$ in $\mathbb{R}^n$ can be identified with the vector $\vec{p}$ from the origin 0 to $p$. The Euclidean distance between $p, q \in \mathbb{R}^n$ is denoted by $|p - q|$, which is the length of the difference vector $\vec{p} - \vec{q}$. For infinite periodic crystals, the theoretically better alternative is the *bottleneck distance* $d_B(S, Q)$ equal to the maximum Euclidean distance needed to perturb every point $p \in S$ to its unique match in $Q$. Formally, $d_B(S, Q) = \inf\limits_{g:S \to Q} \sup\limits_{p \in S} |p - g(p)|$ is minimized over all bijections $g$ and maximized over all points $p \in S$ for a fixed bijection. But $d_B$ is not a metric on all periodic sets because $d_B = +\infty$ for nearly identical lattices, and $d_B$ is discontinuous under perturbations of lattice bases below.

**Example 2.1** (infinite bottleneck). *We show that $S = \mathbb{Z}$ and $Q = (1 + \delta)\mathbb{Z}$ for any $\delta > 0$ have $d_B(S, Q) = +\infty$. Assuming that $d_B(S, Q)$ is finite, consider an interval $[-N, N] \subset \mathbb{R}$ containing $2N + 1$ points of $S$. If there is a bijection $g : S \to Q$ such that $|p - g(p)| \le d_B$ for all points $p \in S$, the image of $2N + 1$ points $S \cap [-N, N]$ under this bijection $g$ should be within the interval $[-N - d_B, N + d_B]$. The last interval contains only $1 + \frac{2(N + d_B)}{1 + \delta}$ points, which is smaller than $1 + 2N$ when $\frac{N + d_B}{1 + \delta} < N$ or $d_B < \delta N$. We get a contradiction by choosing a large $N > \frac{d_B}{\delta}$.* ∎

If we consider only periodic point sets $S, Q \subset \mathbb{R}^n$ with the same density (or unit cells of the same size), the bottleneck distance $d_B(S, Q)$ becomes a well-defined wobbling distance [16].

**Example 2.2** (discontinuity of the wobbling distance). *Slightly perturb the basis $(1, 0), (0, 1)$ of the integer lattice $\mathbb{Z}^2$ to the basis vectors $(1, 0), (\varepsilon, 1)$ of the new lattice $\Lambda$. We prove that $d_B(\Lambda, \mathbb{Z}^2) \ge \frac{1}{3}$ for any $\varepsilon > 0$. Map $\mathbb{R}^2$ by $\mathbb{Z}^2$-translations to the unit square $[0, 1]^2$ with identified opposite sides (a torus). Then the whole square lattice $\mathbb{Z}^2$ is mapped to the single point represented by the corners of the square $[0, 1]^2$. The perturbed lattice $\Lambda$ maps to the sequence of points $\{k\varepsilon \pmod{1}\}_{k=0}^{+\infty} \times \{0, 1\}$ in the horizontal edges. If $d_B(\Lambda, \mathbb{Z}^2) = r < \frac{1}{3}$, then all above points should be covered by the closed disks of the radius $r$ centered at the corners of $[0, 1]^2$. For $0 < \varepsilon < \frac{1}{3} - r$, we can find $k \in \mathbb{Z}$ so that $k\varepsilon$ is strictly between $r, 1 - r$, hence not covered by these disks, so $d_B(\Lambda, \mathbb{Z}_2) \ge \frac{1}{3}$.* ∎

The discontinuity under perturbations is the major weakness of many past invariants including Voronoi diagrams, which should be matched via infinitely many rotations [47], space groups and other group-theoretic invariants [36]. The key example in Fig. 2 (right) shows that a continuous distance between nearly identical sets should be close to 0, not exactly 0. These sets have different symmetries and can be related only by pseudo-symmetries depending on manual thresholds [70].

**Periodic Geometry** develops invariants for Problem 1.1. For any $k \ge 1$, the $k$-th *density function* $\psi_k(t)$ of a periodic set $S = \Lambda + M \subset \mathbb{R}^3$ is the total volume of the regions within the unit cell $U$ of $\Lambda$ covered by exactly $k$ balls $B(p; t)$ with a radius $t \ge 0$ and centres at motif points $p \in M$, divided by the cell volume $\text{Vol}[U]$. The density function $\psi_k(t)$ was proved in [26] to be invariant under isometry, continuous under perturbations, complete for periodic sets satisfying certain conditions of general position in $\mathbb{R}^3$, and computable in a polynomial time in the motif size of $S$ [61]. Section 5 in [26] gave the counter-example to completeness: the 1D sets $S_{15} = X + Y + 15\mathbb{Z}$ and $Q_{15} = X - Y + 15\mathbb{Z}$ for $X = \{0, 4, 9\}$ and $Y = \{0, 1, 3\}$ [39, section 4] have the same density functions for all $k \ge 1$ [3, Example 10] but were distinguished in [67, Example 5b], [5, Example 6.3]. The latest advance [2] reduces Problem 1.1 to an *isoset* of isometry classes of $\alpha$-clusters around points in a motif at a certain radius $\alpha$, which was motivated by the seminal work of Dolbilin [34, 11, 23]. The continuous metrics on isosets [4, section 5] have only an approximate algorithm. Despite the progress in dimension 1 [43] and lattices in dimension 2 [44, 15] and 3 [42, 14], Problem 1.1 remained open in general.

# 3   The Pointwise Distance Distribution of a finite or periodic point set

Distances to neighbors were considered in [67, Definition 5], though only their average was proved to be invariant. Definition 3.1 introduces the matrix PDD whose ordered rows guarantee invariance in Theorem 3.2 and weights of rows that guarantee continuity under perturbations in Theorem 4.3.

**Definition 3.1** (Pointwise Distance Distribution PDD). *For a motif of points $M = \{p_1, \ldots, p_m\}$ in a unit cell $U$ of a lattice $\Lambda$, let $S \subset \mathbb{R}^n$ be a finite set coinciding with $M$ or a periodic set $S = \Lambda + M$. For an integer $k \geq 1$, consider the $m \times k$ matrix $D(S; k)$, whose $i$-th row consists of the ordered distances $d_{i1} \leq \cdots \leq d_{ik}$ measured from $p_i$ to its first $k$ nearest neighbors in the full set $S$. The rows of $D(S; k)$ are lexicographically ordered as follows. A row $(d_{i1}, \ldots, d_{ik})$ is smaller than $(d_{j1}, \ldots, d_{jk})$ if a few first distances coincide: $d_{i1} = d_{j1}, \ldots, d_{il} = d_{jl}$ for $l \in \{1, \ldots, k-1\}$ and the next $(l+1)$-st distances satisfy $d_{i,l+1} < d_{j,l+1}$. If $w$ rows are identical to each other, any such group is collapsed to one row with the* weight $w/m$. *For each row, put this weight in the first column. The final $m \times (k+1)$-matrix is the* Pointwise Distance Distribution PDD$(S; k)$. ∎

The matrix $D(T; 3)$ in Table 1 has two pairs of identical rows, so the matrix PDD$(T; 3)$ consists of two rows of weight $\frac{1}{2}$ below. The matrix $D(K; 3)$ in Table 1 has only one pair of identical rows, so PDD$(K; 3)$ has three rows of weights $\frac{1}{2}, \frac{1}{4}, \frac{1}{4}$. Hence PDD$(T; 3) \neq$ PDD$(K; 3)$.

Table 1: Each point in $T, K \subset \mathbb{R}^2$ from Figure 3 has ordered distances to three other points.

| $T$ points | neighb.1 | neighb.2 | neighb.3 | $K$ points | neighb.1 | neighb.2 | neighb.3 |
|---|---|---|---|---|---|---|---|
| $(-2, 0)$ | $\sqrt{2}$ | $\sqrt{10}$ | $4$ | $(-2, 0)$ | $\sqrt{2}$ | $\sqrt{2}$ | $4$ |
| $(+2, 0)$ | $\sqrt{2}$ | $\sqrt{10}$ | $4$ | $(+2, 0)$ | $\sqrt{10}$ | $\sqrt{10}$ | $4$ |
| $(-1, 1)$ | $\sqrt{2}$ | $2$ | $\sqrt{10}$ | $(-1, -1)$ | $\sqrt{2}$ | $2$ | $\sqrt{10}$ |
| $(+1, 1)$ | $\sqrt{2}$ | $2$ | $\sqrt{10}$ | $(-1, +1)$ | $\sqrt{2}$ | $2$ | $\sqrt{10}$ |

$$\text{PDD}(T; 3) = \begin{pmatrix} 1/2 & \sqrt{2} & 2 & \sqrt{10} \\ 1/2 & \sqrt{2} & \sqrt{10} & 4 \end{pmatrix} \neq \text{PDD}(K; 3) = \begin{pmatrix} 1/4 & \sqrt{2} & \sqrt{2} & 4 \\ 1/2 & \sqrt{2} & 2 & \sqrt{10} \\ 1/4 & \sqrt{10} & \sqrt{10} & 4 \end{pmatrix}.$$

**Theorem 3.2** (invariance of PDD, all theorems are proved in the extended version [66]). *For any finite or periodic set $S \subset \mathbb{R}^n$, PDD$(S; k)$ from Definition 3.1 is an isometry invariant of $S$, $k \geq 1$.* ∎

Table 2: Distances from each motif point of $S(r)$ and $Q(r)$ to their closest neighbors in Fig. 3.

| $S(r)$ points | distance to neighbor 1 | distance to neighbor 2 | distance to neighbor 3 |
|---|---|---|---|
| $p_1 = 0$ | $|0 - r| = r$ | $|0 - (2+r)| = 2+r$ | $|0 - 4| = 4$ |
| $p_2 = r$ | $|r - 0| = r$ | $|r - (2+r)| = 2$ | $|r - 4| = 4 - r$ |
| $p_3 = 2+r$ | $|(2+r) - 4| = 2-r$ | $|(2+r) - r| = 2$ | $|(2+r) - 0| = 2+r$ |
| $p_4 = 4$ | $|4 - (2+r)| = 2-r$ | $|4 - r| = 4-r$ | $|4 - 0| = 4$ |

| $Q(r)$ points | distance to neighbor 1 | distance to neighbor 2 | distance to neighbor 3 |
|---|---|---|---|
| $p_1 = 0$ | $|0 - (2+r)| = 2+r$ | $|0 - (r + 4 - 8)| = 4 - r$ | $|0 - 4| = 4$ |
| $p_2 = 2+r$ | $|(2+r) - 4| = 2-r$ | $|(2+r) - (4+r)| = 2$ | $|(2+r) - 0| = 2+r$ |
| $p_3 = 4$ | $|4 - (4+r)| = r$ | $|4 - (2+r)| = 2-r$ | $|4 - 0| = 4$ |
| $p_4 = 4+r$ | $|(4+r) - 4| = r$ | $|(4+r) - (2+r)| = 2$ | $|(4+r) - 8| = 4-r$ |

For the 1D periodic sets $S(r) = \{0, r, 2+r, 4\} + 8\mathbb{Z}$ and $Q(r) = \{0, 2+r, 4, 4+r\} + 8\mathbb{Z}$ with unit cell $[0, 8]$ in Fig. 3, Table 2 shows that $S(r), Q(r)$ are not isometric for any parameter $0 < r \leq 1$.

$$\text{PDD}(S(r); 8) = \begin{pmatrix} 1/4 & r & 2+r & 4 & 4 & 6-r & 8-r & 8 & 8 \\ 1/4 & r & 2 & 4-r & 4+r & 6 & 8-r & 8 & 8 \\ 1/4 & 2-r & 2 & 2+r & 6-r & 6 & 6+r & 8 & 8 \\ 1/4 & 2-r & 4-r & 4 & 4 & 4+r & 6+r & 8 & 8 \end{pmatrix} \neq$$

$$\text{PDD}(Q(r); 8) = \begin{pmatrix} 1/4 & r & 2-r & 4 & 4 & 6+r & 8-r & 8 & 8 \\ 1/4 & r & 2 & 4-r & 4+r & 6 & 8-r & 8 & 8 \\ 1/4 & 2-r & 2 & 2+r & 6-r & 6 & 6+r & 8 & 8 \\ 1/4 & 2+r & 4-r & 4 & 4 & 4+r & 6-r & 8 & 8 \end{pmatrix}.$$

The averages of columns in $\mathrm{PDD}(S;k)$ form the vector $\mathrm{AMD}(S;k)$ of *Average Minimum Distances* [67, Definition 5]. Any lattice $\Lambda \subset \mathbb{R}^n$ has a 1-point motif $M = \{p\}$, hence $\mathrm{PDD}(S;k)$ is a single row of increasing distances from $p$ to all other points $\Lambda - \{p\}$. Fig. 6 (right) shows a honeycomb set $S$ whose motif consists of two points that have the same distances to all their neighbors, hence two rows of $D(S;k)$ collapse to a single vector. So $\mathrm{PDD}(S;k) = \mathrm{AMD}(S;k)$ in the cases above.

**Example 3.3** (PDD is stronger than AMD). *Since both sets $S(r), Q(r)$ in Fig. 3 (right) have period 8, the matrices $\mathrm{PDD}(S(r);k)$ and $\mathrm{PDD}(Q(r);k)$ have distance 8 in each row for columns 7 and 8 as shown above. All further distances are obtained from the first eight by adding a multiple of period 8. The vector $\mathrm{AMD}(S(r);k)$ of column averages for any $k \geq 8$ is determined by $\mathrm{AMD}(S(r);8) = (1, 2.5, 3.5, 4.5, 5.5, 7, 8, 8)$. Since the components of $\mathrm{AMD}(S(r);k)$ do not depend on the parameter $0 < r < 1$, the sets $S(r)$ are counter-examples to the completeness of $\mathrm{AMD}$, now distinguished by $\mathrm{PDD}(S(r);k)$ already for $k = 1$. Hence $\mathrm{PDD}(S;k)$ is strictly stronger than $\mathrm{AMD}(S;k)$.* ∎

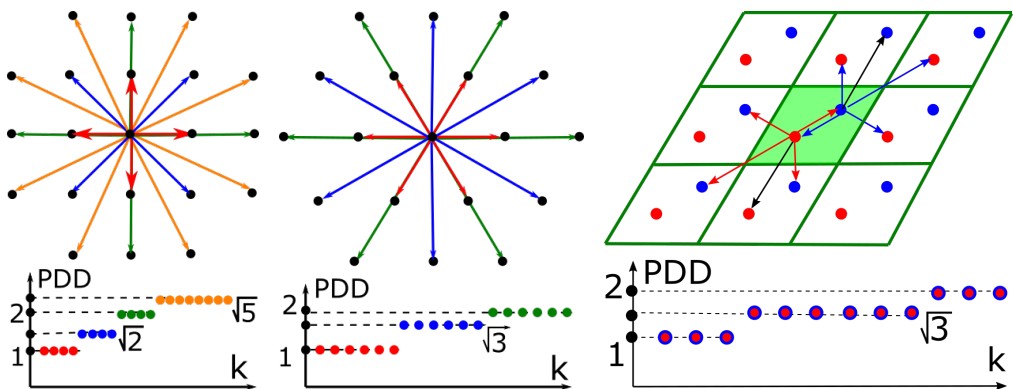

Figure 6: The square lattice (left), hexagonal lattice (middle), and honeycomb periodic set (right) with a minimum inter-point distance of 1 have $\mathrm{PDD}(S;k)$ with a single row of increasing distances.

For a periodic set $S$, the number $k$ in $\mathrm{PDD}(S;k)$ can be considered as a degree of approximation (or a count of decimal places), not as a parameter that affects invariant values. If we increase $k$, we extract more distant geometric data from $S$ by adding more columns to $\mathrm{PDD}(S;k)$ and keeping all previous distances. If some rows are identical in $D(S;k-1)$ and become different in $D(S;k)$, we recompute weights but not distances. The past tools [18, 7] strongly depend on extra parameters.

Now we compare PDD with the closest past invariant, the Pair Distribution Function (PDF). For a periodic point set $S \subset \mathbb{R}^n$ with a motif $M$, the *exact* $\mathrm{ePDF}(S)$ consists of ordered distances from all points $p \in M$ to all other points $q \in S - \{p\}$. So the infinite sequence $\mathrm{ePDF}(S)$ is obtained from PDD by combining all rows into one sequence and losing weights. Additionally, we keep only one distance from each pair $|p - q| = |q - p|$. For any fixed $0 < r \leq 1$, the sets $S(r), Q(r)$ have the same sequences starting with $\mathrm{ePDF}(S(r)) = \mathrm{ePDF}(Q(r)) = \{r, 2 - r, 2, 2 + r, 4 - r, 4, 4, 4 + r, \dots\}$. This example shows that PDD for $k = 1$ is strictly stronger than ePDF as an isometry invariant.

For any lattice $\Lambda \subset \mathbb{R}^n$, the vector $\mathrm{ePDF}(\Lambda;k)$ up to $k$ distances concides with $\mathrm{PDD}(S;k)$. For the honeycomb set $S$ in Fig. 6 (right), $\mathrm{ePDF}(S;2k)$ is obtained from $\mathrm{PDD}(S;k)$ by repeating every distance twice. If a periodic set is perturbed and a unit cell doubles as in Fig. 2 (right), then every distance in ePDF is replaced by a couple of (near-) duplicate distances, so $\mathrm{ePDF}(S;k)$ discontinuously changes by including twice as many short distances and losing longer distances.

This typical discontinuity was roughly repaired by replacing every single distance $d$ with its Gaussian distribution $\exp(-(x - d)^2 / 2\sigma)$ with a parameter $\sigma > 0$. Then a normalized sum of such 'blurred' distances [63] becomes the smooth Pair Distribution Function $\mathrm{PDF}(S;\sigma)$. Since algorithms can compare only finite vectors, this $\mathrm{PDF}(S;\sigma)$ is then uniformly sampled, which creates dependence on $\sigma$ and sampling. So PDD provides an easy alternative to this counter-intuitive PDF pipeline {discrete sequence} $\rightarrow$ {smooth function} $\rightarrow$ {discrete sequence}, whose continuity wasn't formally proved.

# 4 Continuity and generic completeness of Pointwise Distance Distributions

Continuity of $\text{PDD}(S; k)$ under perturbations of $S$ in the bottleneck distance $d_B$ will be measured by the Earth Mover's Distance [55], which can be applied to any weighted distributions of different sizes. Definition 4.1 is for any vector $I(S) = ([w_1(S), R_1(S)], \ldots, [w_{m(S)}, R_{m(S)}(S)])$ of pointwise invariants of a set $S$ with weights $w_i(S) \in (0, 1]$ satisfying the normalization $\sum_{i=1}^{m(S)} w_i(S) = 1$.

Later we consider only the case when the weighted vector $[w_i, R_i]$ is the $i$-th row of $\text{PDD}(S; k)$. Then $m(S)$ is the number of rows in $\text{PDD}(S; k)$. Each row $R_i(S)$ should have a size independent of $S$, for example a number $k$ of neighbors in $\text{PDD}(S; k)$. For any vectors $R_i = (r_{i1}, \ldots, r_{ik})$ and $R_j = (r_{j1}, \ldots, r_{jk})$ of a length $k$, we use the $L_\infty$-distance $|R_i - R_j|_\infty = \max_{l=1,\ldots,k} |r_{il} - r_{jl}|_\infty$.

**Definition 4.1** (EMD). *Let finite or periodic sets $S, Q \subset \mathbb{R}^n$ have weighted vectors $I(S), I(Q)$ as discussed above. A flow from $I(S)$ to $I(Q)$ is an $m(S) \times m(Q)$ matrix whose element $f_{ij} \in [0, 1]$ represents a partial flow from $R_i(S)$ to $R_j(Q)$. The Earth Mover's Distance is the minimum cost*

$$\text{EMD}(I(S), I(Q)) = \sum_{i=1}^{m(S)} \sum_{j=1}^{m(Q)} f_{ij} |R_i(S) - R_j(Q)| \text{ for } f_{ij} \in [0, 1] \text{ subject to } \sum_{j=1}^{m(Q)} f_{ij} \leq w_i(S)$$

*for $i = 1, \ldots, m(S)$, $\sum_{i=1}^{m(S)} f_{ij} \leq w_j(Q)$ for $j = 1, \ldots, m(Q)$, and $\sum_{i=1}^{m(S)} \sum_{j=1}^{m(Q)} f_{ij} = 1$.* ∎

The first condition $\sum_{j=1}^{m(Q)} f_{ij} \leq w_i(S)$ means that not more than the weight $w_i(S)$ of the component $R_i(S)$ 'flows' into all components $R_j(Q)$ via 'flows' $f_{ij}$, $j = 1, \ldots, m(Q)$. Similarly, the second condition $\sum_{i=1}^{m(S)} f_{ij} = w_j(Q)$ means that all 'flows' $f_{ij}$ from $R_i(S)$ for $i = 1, \ldots, m(S)$ 'flow' into $R_j(Q)$ up to the maximum weight $w_j(Q)$. The last condition $\sum_{i=1}^{m(S)} \sum_{j=1}^{m(Q)} f_{ij} = 1$ forces to 'flow' all rows $R_i(S)$ to all rows $R_j(Q)$. The EMD satisfies all metric axioms [55, appendix], needs $O(m^3 \log m)$ time for distributions of a maximum size $m$ and is approximated in $O(m)$ time [59, 57].

**Theorem 4.2** (lower bound of EMD). *For finite or periodic point sets $S, Q \subset \mathbb{R}^n$, and $k \geq 1$, the distances satisfy $\text{EMD}(\text{PDD}(S; k), \text{PDD}(Q; k)) \geq ||\text{AMD}(S; k) - \text{AMD}(Q; k))||_\infty$.* ∎

Theorem 4.3 uses the bottleneck distance $d_B(S, Q) = \inf_{g: S \to Q} \sup_{p \in S} |p - g(p)|$ and the *packing radius* $r(S)$, which is the minimum half-distance between any points of $S$. Equivalently, $r(S)$ is the maximum radius $r$ to have disjoint open balls of radius $r$ centered at all points of $S$.

**Theorem 4.3** (continuity of PDD). *For any $k \geq 1$, if finite or periodic sets $S, Q \subset \mathbb{R}^n$ satisfy $d_B(S, Q) < r(S)$, then $\text{EMD}(\text{PDD}(S; k), \text{PDD}(Q; k)) \leq 2d_B(S, Q)$.* ∎

Continuity Theorem 4.3 means that any small perturbation of atomic positions in the bottleneck distance $d_B$ leads to a small change of the Pointwise Distance Distribution in the Earth Mover's Distance. Theorem 4.3 extends the following fact for 2-point sets ($k = 1$). If we perturb two points by at most $\varepsilon$, the distance between them changes by at most $2\varepsilon$.

For any set $S \subset \mathbb{R}^n$ of $m$ points with distinct inter-point distances, completeness of $\text{PDD}(S; m - 1)$ follows from [67, Theorem 16]. Following the earlier work [26, section 5.1], the extended version [66] defines a distance-generic set that can approximate any periodic point set $S = \Lambda + M \subset \mathbb{R}^n$. The number $m$ of points in a unit cell $U$ is an isometry invariant because any isometry maps $U$ to another cell with the same number $m$ of points. Theorem 4.4 assumes that a lattice $\Lambda$ is given and reconstructs a periodic point set $S = \Lambda + M$ in any dimension $n \geq 2$. However, in low dimensions $n = 2, 3$, any lattice $\Lambda$ can be reconstructed from its isometry invariants described in [19, 44, 42].

**Theorem 4.4** (generic completeness of PDD). *Let $S = \Lambda + M \subset \mathbb{R}^n$ be a distance-generic periodic set with $m$ points in a motif $M$. Let $R(\Lambda)$ be the smallest radius such that all closed balls with centers $p \in \Lambda$ cover $\mathbb{R}^n$. Let $2R(\Lambda)$ be smaller than all distances in the last column of $\text{PDD}(S; k)$ for a big enough $k$. The set $S$ can be uniquely reconstructed up to isometry from $\Lambda$, $m$, and $\text{PDD}(S; k)$.* ∎

# 5 Polynomial time algorithms and experimental comparisons of PDD

The algorithm for PDD in Theorem 5.1 found several pairs of unexpected duplicates, which were missed by all past tools, through 200B+ pairwise comparisons of 660K+ real periodic crystals over a couple of days on AMD Ryzen 5 5600X (6-core) @4.60Ghz, 32GB DDR4 RAM @3600 Mhz.

The main input is a periodic point set $S \subset \mathbb{R}^n$ given by a unit cell $U$ and a motif of $m$ points in $U$. The key parameters of $\mathrm{PDD}(S; k)$ are $m$ and $k$. The complexity in Theorem 5.1 is near-linear in both $k$ and $m$ for a fixed dimension $n$. The output $\mathrm{PDD}(S; k)$ is a matrix with at most $m$ rows and exactly $k + 1$ columns. The first column contains the weights of rows, which sum to 1 and are proportional to the number of appearances of the row before collapsing, see the detailed code in [66].

**Theorem 5.1** (PDD complexity). *Let a periodic set $S \subset \mathbb{R}^n$ have $m$ points in a unit cell $U$. For a fixed dimension $n$, $\mathrm{PDD}(S; k)$ is computed in a near-linear time $O(km(5\nu)^n V_n \log(m) \log^2(k))$, where $V_n$ is the unit ball volume in $\mathbb{R}^n$, $d$ and $\nu = \frac{d}{\sqrt[n]{\mathrm{Vol}[U]}}$ are the diameter and* skewness *of $U$.* ∎

The near-linear time complexity in Theorem 5.1 goes far beyond the state-of-the-art. Section 2 reviewed that all past tools are based on ambiguous non-invariant data or discontinuous invariants that miss (near-)duplicates, or are too slow for pairwise comparisons of millions of crystal structures. The recent continuous invariants with theoretical (not exactly computable) metrics [26, section 6] and [1, section 8] require cubic algorithms, which turned out to be unrealistic for large crystal datasets.

The Cambridge Crystallographic Data Centre (CCDC) has curated the world's largest collection of real solid crystalline materials since the 1960s, now having nearly 1.2M known structures. New crystalline material is deposited in the CSD only after a peer-reviewed publication. The CCDC should ideally check that a new structure is genuine and not a duplicate of an earlier one because their data is trusted by all pharmaceutical giants developing new drugs in a crystalline form. However, the CSD is basically a huge list of Crystallographic Information Files representing crystals by unit cells and motifs of points in coordinates of a cell basis with limited search and comparison capabilities.

The papers [29, 69] reported five experimental T2 crystals (based on the same molecule T2) that were successfully synthesized after predicting 5679 crystals through 12-week simulations on a supercomputer. All initial 2M+ randomly sampled crystals were iteratively optimized to the 'most stable' approximations of local energy minima. This is a typical 'embarrassment of over-prediction' when many (near-)duplicates are found around the same local minimum but remain undetected.

One striking example is the pair of crystals 14 and 15 in Fig. 7 reported in [29], which have very different unit cells but are nearly identical. When this pair was compared by another free software Platon [62], a bug was discovered, which required a few months to be fixed. Such bugs will keep emerging because the discontinuity of past invariants and metrics was not addressed as in Problem 1.1.

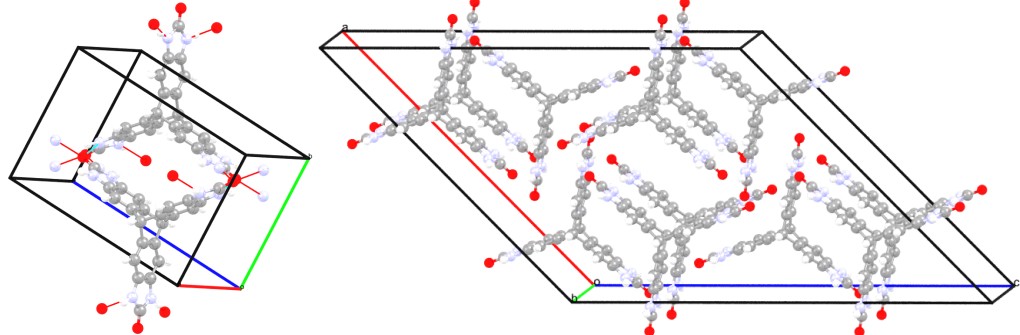

Figure 7: Crystals 14, 15 based on the T2 molecule have very different Crystallographic Information Files (with different motifs in unit cells of distinct shapes) but are nearly identical up to isometry.

For example, a rough sampling of the density functions $\psi_k(t)$ from [26] of 5679 crystals for up to $k = 8$ took more than four days on a comparable machine. This experiment detected the T2-$\delta$ crystal that was accidentally not deposited in the CSD because of visual confusion with another structure.

# 6 Conclusions and discussions of current limitations and duplicate detection

The popular packing similarity [18] algorithm COMPACK is available in the CCDC's Mercury software. The 4950 comparisons of the 100 lowest energy crystals close to T2-$\delta$ in density by packing similarity took 3 hours 53 min, 2.8 seconds. Extrapolating this time for comparing any new structure with the whole CSD gives 38 days, hence more than 34 thousand years for pairwise comparisons of all 660K+ periodic crystals in the CSD. A typical comparison by AMD and PDD runs in nanoseconds and milliseconds, respectively, so comparing 100 crystals pairwise takes less than one minute.

Table 3: Most comparisons of 100 lowest energy crystals close to the T2-$\delta$ by packing similarity [18] matched small numbers of molecules for the default maximum 15, which means a failure to match.

| molecules | 1 | 2 | 3 | 4 | 5 | 6 | 7 | 8 | 9 | 10 | 11 | 12 | 13 | 14 | 15 |
|---|---|---|---|---|---|---|---|---|---|---|---|---|---|---|---|
| comparisons | 2784 | 1150 | 773 | 69 | 85 | 21 | 31 | 6 | 2 | 7 | 12 | 3 | 1 | 0 | 6 |

Most importantly, more than half of all comparisons in Table 3 matched only one molecule from two crystals. Since all crystals consist of the same rigid molecule T2, this output means a complete failure: one common molecule, no other conclusions. Since the CCDC deposits hundreds of new structures daily, the shortcut approach is to compare the chemical composition of atoms. But even water ($H_2O$) has at least 15 forms of ice crystals, while other compositions have many more polymorphic forms in the CSD. This comparison by composition can miss duplicates where one atom is incorrect.

The unreliability of past comparisons fueled the growing crisis of fake crystallographic data [10] leading to the investigation of nearly 1000 structures in the CSD. The new invariants PDD detected five pairs of unexpected duplicates ('needles in the haystack') in the Cambridge Structural Database (CSD). First, the simpler invariants $AMD(S; 100)$ were computed for all 660K+ periodic structures in the CSD, without disorder and with full geometric data. More than 200 billion pairwise comparisons of $AMD(S; 100)$ vectors revealed 6371 pairs $S, Q$ with $|AMD(S; 100) - AMD(Q; 100)|_\infty \leq 0.01$. As an AMD is simpler and faster to compare, up to the order of $10^{-7}$ seconds per comparison, this step took around 8 hours. This example and Theorem 4.2 show AMD is a good filter.

Second, computing the $L_\infty$-based EMD between the pairs above detected 182 pairs with EMD $<$ 0.01. Most of these pairs were expected and were the same crystal, or different aliases for the same database entry. The five pairs reported in [67, section 7] were unexpected because the underlying periodic sets of points at atomic centres were truly isometric (to the last decimal place) but one atom had different chemical elements in two crystals. The crystals with the CSD codes HIFCAB and JEPLIA are literally isometric but one Cadmium is replaced by Manganese at the same position.

All past tools taking into account atomic types see these geometric duplicates as different. The CCDC agreed that such a coincidence is physically impossible because another atom should have slightly different distances to neighbors detected by PDD. Hence at least one of the structures in each pair cannot be correct. The five journals are investigating the data integrity of the underlying publications. Detecting duplicates might have some negative impact on past publications that could be retracted

In conclusion, Theorems 3.2, 4.3, 5.1 , 4.4 fulfilled almost all conditions of Problem 1.1, while all past tools remained discontinuous or too slow for billions of comparisons. The only limitation is a hypothetical existence of singular sets $S \not\cong Q$ with $PDD(S; k) = PDD(Q; k)$ for all $k \geq 1$. We found no such examples among all real materials in the CSD. Even if such a singular pair of $S, Q$ emerges, they can be distinguished by the slower and provably complete invariant isoset [2, 4].

Hence PDD and complete isoset can be considered as a DNA-style code or materials genome uniquely and continuously identifying any periodic crystal. More importantly, the experiments confirmed the *Crystal Isometry Principle* [67, section 7]: the map {periodic crystals} $\rightarrow$ {periodic point sets} is injective (doesn't lose information) modulo isometry. Hence all existing and not yet undiscovered crystals live in the common *Crystal Isometry Space* (CRISP) of isometry classes of periodic point sets continuously parameterized by complete isometry invariants. This CRISP can be considered a continuous universe containing all known crystals as visible stars in the universe, which requires mapping and exploration. Since inter-atomic distances cannot be arbitrary, the next step is to describe subspaces of CRISP that represent realistic but still unknown materials. This first map of this space for 2D lattices appeared in [15]. We thank all reviewers for their valuable time and suggestions.

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
