# OpenReview forum: "Resolving the data ambiguity for periodic crystals"
_NeurIPS.cc/2022/Conference — NeurIPS 2022 Accept_

### Official Review · Reviewer_wsDJ · 2022-07-09

**Rating:** 5
**Confidence:** 3
**Soundness:** 3 good
**Presentation:** 3 good
**Contribution:** 2 fair

**Summary:**

This paper focuses on the problem of geometric isomorphism for periodic structures of materials. They propose a new crystal descriptor without false negatives, which means if two periodic structures are the same, the output of this PDD descriptor will be the same. This PDD descriptor has a near-linear time complexity. They achieve this by formulating a PDD matrix using k-nearest pairwise distances for all atoms in a unit cell.

**Questions:**

My main concerns are the weaknesses 2 and 3 mentioned above, could you please give some clarifications about them?

**Limitations:**

Yes.

**Strengths And Weaknesses:**

## Strengths
* This proposed descriptor has near-linear time-complexity without false negative when only the positions of points considered.
* The illustration of different unit cell representations for the same material is insightful and easy to understand.
* This proposed method can deal with a larger unit cell representation as shown in Fig.2 in the main paper.

## Weaknesses
* Originality: The idea of using k-nearest pairwise distances is not new. CGCNN proposed by Xie et al. actually uses k-nearest pairwise distances to form the crystal graph representation to make the graph input invariant to different unit cell representations.
* The false positive, which is defined in 1.1b, can not be satisfied when the k is given (large enough k can not be achieved during usage of this PDD descriptor). The increase of k will  increase the time-complexity at least linearly.
* The false negative, which is defined in 1.1a, can not be satisfied when you consider atomic numbers of points. Two different materials with the same geometric structure and a bit different atomic numbers will result in the same output of PDD descriptor.

---

> ### Author Response · Authors · 2022-08-02
> **Crystal graphs are known to be discontinuous under perturbations. The new continuous invariants distinguish all real periodic crystals even without atomic types.**
>
> Thank you for the review.
>
> >The idea of using k-nearest pairwise distances is not new. CGCNN proposed by Xie et al.
>
> CGCCN by Xie uses thresholds to define edges between atomic vertices, see their supplementary materials: “6 Angstrom radius … interatomic distance lower than the sum of the Cordero covalent bond lengths with a 0.25A tolerance”.  Any combinatorial graph is discontinuous under perturbations of atoms and cannot reliably identify nearly identical crystals. Any fixed number k of neighbors (k=12 in the above paper) similarly leads to discontinuity, e.g. the Voronoi domain of a rectangular 2D lattice has four vertices (or Voronoi neighbors) but any generic perturbation gives a general lattice whose Voronoi domain is hexagonal with six vertices.
>
> This discontinuity is widely known. Google Scholar reports many publications for “crystal graph discontinuity”. So the novelty is not in using k-nearest neighbors but in combining continuous distances into PDD matrices that are continuous under perturbations in the Earth Mover’s Distance, which is a continuous metric on weighted distributions of PDD rows.
>
> >The false positive, which is defined in 1.1b, can not be satisfied when the k is given (large enough k can not be achieved during usage of this PDD descriptor).
>
> Theorem 4.4 provides a small upper bound for k via the packing radius that can be computed in advance for a given periodic point set.
>
> >The increase of k will increase the time-complexity at least linearly.
>
> Yes, this near-linear dependence on k and the number m of motif points is the key result that allowed more than 200 billion pairwise comparisons of all 660K+ real periodic crystals within two days on a modest desktop.
>
> >The false negative, which is defined in 1.1a, can not be satisfied when you consider atomic numbers of points. Two different materials with the same geometric structure and a bit different atomic numbers will result in the same output of PDD descriptor.
>
> The key discovery says that there are *no different real periodic crystals* with the same geometric structure!
>
> This Crystal Isometry Principle is justified by the above experiment on the Cambridge Structural Database. Indeed, all crystallographers who looked at our detected duplicates (missed by all past tools) politely called them suspicious. The Cambridge Crystallographic Data Centre agreed and initiated investigations with five journals.
>
> These duplicates are physically impossible because replacing one atom with another one (for example, Cd with Mn having atomic numbers 25 and 48 in the pair HIFCAB vs JEPLIA) should change distances to neighbors at least slightly, which is immediately detected by our distance-based invariants.
>
> All these duplicates are under investigation by five journals, which will likely lead to retractions of publications with falsified data.
>
> In other words, this detection of duplicates is a practical example of finding five pairs of *identical needles in a haystack*. The Cambridge Structural Database is a huge collection of ambiguous Crystallographic Information Files, which cannot be compared quickly enough. Even worse, all past similarity measures fail the triangle axiom of a metric or are discontinuous under perturbations.
>
> Hence this paper shows a classical example of how a hard Data Science problem can be efficiently solved, not by brute force black box tools, but by rigorous methods and mathematical proofs.
>
> The huge consequence of the experimental fact above is that all periodic crystals live in a common space of isometry classes of all periodic sets of points without atomic types.
>
> This Crystal Isometry Space (CRISP) provides a map of the crystal universe at infinite resolution with all known crystals visible as starts, while all undiscovered crystals are waiting to be found by exploring the new map.
>
> The map of CRISP is continuously infinite and is parameterized by complete isometry invariants, for example, isosets from ref [3], though the generically PDD invariants are much faster to compute.
>
> The parameterization of CRISP by continuous invariants is like mapping a surface of a new planet by geographic style coordinates whose values uniquely determined the location of any crystal in the map of CRISP. All past approaches studied this continuous space of crystals either by discrete tools (symmetry groups define low-dimensional subspaces) or by blindly sampling the previously unmapped space without understanding which regions of CRISP were explored and which are not.
>
> The absence of chemical labels assigned to points is the key advantage because fixing a chemical composition splits the continuous CRISP into a huge variety of unrelated small subspaces.
>
> Like Mendeleev’s table combining all chemical elements (despite their differences) into a single table parameterized by two discrete coordinates (period and group number), the CRISP puts all periodic crystals into a common continuous space parameterized by invariants such as PDD.

---

> > ### Comment · Reviewer_wsDJ · 2022-08-03
> > **Still some concerns**
> >
> > > The false positive, which is defined in 1.1b, can not be satisfied when the k is given (large enough k can not be achieved during usage of this PDD descriptor).
> >
> > I am not fully convinced, how to compute a small upper bound for k via the packing radius? Theorem 4.4 is not about the upper bound.
> >
> > > The false negative, which is defined in 1.1a, can not be satisfied when you consider atomic numbers of points. Two different materials with the same geometric structure and a bit different atomic numbers will result in the same output of PDD descriptor.
> >
> > You said that 'The key discovery says that there are no different real periodic crystals with the same geometric structure!'. Is this claim a fact? Can you give me a reference?

---

> > > ### Author Response · Authors · 2022-08-03
> > > **The upper bound is described in Theorem 4.4, the reference to the Crystal Isometry Principle is the past work [20]**
> > >
> > > Thank you for your questions.
> > >
> > > >how to compute a small upper bound for k via the packing radius? Theorem 4.4 is not about the upper bound.
> > >
> > > Here is the quote from Theorem 4.4 in lines 243-245 of the submitted paper: “Let R(Λ) be the smallest radius such that all closed balls with centers p in Λ cover the space R^n. Let 2R(Λ) be smaller than all distances in the last column of PDD(S; k) for a big enough k.”
> > >
> > > The radius R(Λ) can be computed from a given basis of the lattice Λ. For example, if v_1,…,v_n is a basis of the lattice Λ, a simple upper bound for this radius R(Λ) is the half-length of a longest vector c_1 v_1 + … + c_n v_n, where coefficients c_1,…,c_n take only values -1,0,+1. In dimension n=2, this upper bound is a half of the maximum between the lengths |v_1|,|v_2|,|v_1+v_2|,|v_1-v_2|.
> > >
> > > Then Theorem 4.4 says that it is enough to compute the matrix PDD(S;k) up to k columns so that the distances in each row of PDD(S;k) go above the double radius 2R(Λ). Then we search for sufficiently many k neighbors (for each point in a unit cell) whose distances should reach the upper bound above.
> > >
> > > > 'The key discovery says that there are no different real periodic crystals with the same geometric structure!'. Is this claim a fact? Can you give me a reference?
> > >
> > > Yes, this fact is referenced in lines 322-324 in the conclusions (section 6) of the submitted paper: “More importantly, the experiments confirmed the Crystal Isometry Principle [20, section 7]: the map: {periodic crystals} -> {periodic point sets} is injective (doesn’t lose information) modulo isometry.”
> > >
> > > So the reference for the established fact that any different (non-isometric) real periodic crystals have non-isometric sets of atomic centers (without chemical elements) is section 7 in the past published work [20]: MATCH Communications in Mathematical and in Computer Chemistry (https://match.pmf.kg.ac.rs), 87:529–559, 2022, available at https://doi.org/10.46793/match.87-3.529W
> > >
> > > This discovery has a simple intuitive explanation at the level of school physics: if one atom in a crystal is a replaced by a different one (say, by a larger atom with more neutrons or protons), then inter-atomic forces should inevitably perturb distances from a new atom to old neighbors, which is immediately captured by our distance-based invariant PDD.
> > >
> > > The physical intuition above is now confirmed by more than 200 billion pairwise comparisons of all (more than 660 thousand) real periodic crystals (without disorder and with full geometric data) in the world’s largest database of solid crystalline materials. Despite this Cambridge Structural Database is well-curated and widely used by all pharmaceutical giants, there was no reliable tool to compare if any newly deposited crystal is really new, not a slight perturbation of another already deposited material.
> > >
> > > The available comparison on a single crystal with all others in the CSD via the Root Mean Square Deviation (RMSD) of atomic displacements (limited to a manually chosen number of atoms or molecules) requires the estimated 38 days on our machine, see line 286 in the submitted paper, while all pairwise comparisons of the new PDD invariants required only two days on the same modest desktop computer.
> > >
> > > More importantly, this RMSD and all other similarity measures, for example, based on traditional X-ray diffraction patterns, do not satisfy the triangle axiom of a metric, see line 117 in the submitted paper.
> > >
> > > Hence, the proposed PDD invariant is the first rigorously justified tool for reliable comparisons of all periodic crystals independent of chemical compositions. The PDD is fast even for pairwise comparisons to detect (near-)duplicates in all experimental databases and to fight the growing crisis of fake data.
> > >
> > > The five previously detected duplicates in [20, section 7] and the nine new pairs in Table 6 of the appendix in the submitted paper are typical examples of finding “identical needles in a haystack” because the Cambridge Structural Database keeps all materials in conventional Crystallographic Information Files (CIFs) that are too ambiguous (any crystal can have infinitely many CIFs) without fast continuous metrics.
> > >
> > > The continuous paramterization of the Crystal Isometry Space (CRISP) can be compared with Mendeleev's table of all chemical elements parameterized by only two discrete coordinates (period and group number), while CRISP includes all known (660K+) periodic crystals parameterized by more continuous (geographic-style) coordinates, which are elements of the PDD matrix.
> > >
> > > The final two lines (325-326) of the submitted paper reference such explicit geographic style maps already constructed for 2D lattices: "Hence all existing and undiscovered crystals live in the common space parameterized by complete isometry invariants. Its first continuous maps for 2D lattices appeared in [9]." Such a map will guide search for new materials instead of blind sampling.
> > >
> > > If anything remains unclear, we would be happy to help.

---

> > > > ### Comment · Reviewer_wsDJ · 2022-08-04
> > > > **Thanks for the response**
> > > >
> > > > I'm a little more convinced.
> > > >
> > > > Also, could you give me an example how you can completely reconstruct the crystal structure from m, lattice and PDD matrix? Also, an example of how to determine the smallest k for several crystals will be appreciated. It seems to me that the geometric structure of the unit cell structure is determined, but there is still the freedom of rotation for unit cell structures.

---

> > > > > ### Author Response · Authors · 2022-08-04
> > > > > **The detailed proofs of all theorems are in appendix C, the key ideas are briefly summarized below.**
> > > > >
> > > > > Thank you for the questions.
> > > > >
> > > > > >an example of how to determine the smallest k for several crystals will be appreciated.
> > > > >
> > > > > For any finite dataset of crystals, a suitable number k of sufficiently many neighbors can be the maximum of these numbers k for each individual crystal. For the Cambridge Structural Database, k=100 was enough to distinguish all 660K+ real periodic crystals by their invariants PDD(S;100).
> > > > >
> > > > > >the geometric structure of the unit cell structure is determined, but there is still the freedom of rotation for unit cell structures.
> > > > >
> > > > > The invariance of the PDD matrix under translations, rotations, and reflections is proved in Theorem 3.2, see lines 165-166 in the submitted paper, the detailed proof is on page 20 of Appendix C in the supplementary materials. Briefly, since distances between points are preserved by any isometry (a composition of translations, rotations, reflections) by definition, the PDD matrix consisting of inter-point distances remains invariant under isometry.
> > > > >
> > > > > > could you give me an example how you can completely reconstruct the crystal structure from m, lattice and PDD matrix?
> > > > >
> > > > > The full proof of this Reconstruction Theorem 4.4 is on pages 22-23 of Appendix C in the supplementary materials.
> > > > > The idea is to uniquely locate every point in a motif by its shortest distances to lattice nodes.
> > > > >
> > > > > For instance, if a 2D lattice Λ has a generic basis of vectors v_1, v_2, then it suffices to reconstruct all points of a periodic set S within a hexagonal Voronoi domain V(Λ;0) whose translation copies tile R^2. Any such point q has its closest neighbor at the origin 0 and two further neighbors at distances d_1(q), d_2(q) among lattice nodes. Definition C.5 of general position (on page 22 in Appendix C) guarantees that three circles with the radii |q|, d_1(q), d_2(q) and centers 0,p_1,p_2 (for lattices nodes p_1,p_2 in the neighbor set N(Λ) from Definition C.3) meet at a single point for a unique choice of d_1(q),d_2(q) and p_1,p_2 among all finitely many choices, where d_1(q),d_2(q) appear in the row of q in PDD(S;k).
> > > > >
> > > > > For example, if a periodic point set S has only one extra point q inside the triangle on the vertices 0, v_1, v_2, then its row in the PDD matrix has distances |q|, |v_1-q|, |v_2-q|, |v_1+v_2-q|, possibly in another order. The general position guarantees that only three circles with the radii |p-q| and centers p (for three choices of p among 0, v_1, v_2, v_1+v_2) intersect at a single point, so this unique intersection should be q.
> > > > >
> > > > > This reconstruction algorithm has been implemented and tested on real crystals also in dimension 3.

---

> ### Comment · Reviewer_wsDJ · 2022-08-05
> **Concerns partially addressed**
>
> My concerns are partially addressed, therefore, I will increase my score a little.

---

> > ### Author Response · Authors · 2022-08-06
> > **All provided responses quoted the original submission.**
> >
> > >My concerns are partially addressed, therefore, I will increase my score a little.
> >
> > Thank you for increasing the score.
> >
> > What concerns were not addressed?

---

### Official Review · Reviewer_TtwX · 2022-07-11

**Rating:** 5
**Confidence:** 3
**Soundness:** 4 excellent
**Presentation:** 3 good
**Contribution:** 3 good

**Summary:**

The authors provide a periodic-invariant feature called Pointwise Distance Distributions (PDD). The feature is constructed by distances to the first k nearest neighbors of every point in a periodic set with near-linear complexity. The authors also prove that this feature can remove ambiguity due to different representations of a crystal cell, is invariant that different basis of the same material won’t change output, and is continuous that small perturbation won’t largely change the output. The authors use this feature to remove duplicate materials and perform their experiments on the Cambridge Structural Database. The effectiveness of the method is shown by less time and fewer failures.

**Questions:**

Figure 3 Left: Why can’t we distinguish between these two graphs? The distance matrices of them are different.

**Limitations:**

The method is not novel enough. I think other methods such as the radius graph or the combination of radius graph and k-NN graph also suffice to distinguish materials.

For experiments, I recommend using PDD for other material databases to remove variance of your experiments; and also using PDD as features of other important tasks such as material energy prediction.


**Strengths And Weaknesses:**

Strength:
The authors give many examples to show problems of periodic representation and the proof of continuity and completeness of PDD is well-stated.

Weakness:
Using the k-nearest distances to build the features is not novel enough. It’s easy to build a crystal graph with k-nearest neighbors. The experiment is only done in one database and the task is not significant enough.

---

> ### Author Response · Authors · 2022-08-02
> **The key evaluation is via mathematical proofs illustrated on the world's largest and most trusted database of 1.17M+ of curated materials.**
>
> Thank you for the detailed review and useful comments.
>
> >The effectiveness of the method is shown by less time and fewer failures.
>
> No failures at all. Invariance Theorem 3.2 proves that PDD is an isometry invariant, hence has *no false negatives*.
>
> >Using the k-nearest distances to build the features is not novel enough. It’s easy to build a crystal graph with k-nearest neighbors.
>
> Any crystal graph is discontinuous under perturbations for the reasons below. If edges are chosen by a distance threshold, they (dis)appear when a threshold is crossed. If edges are chosen by a fixed number of neighbors, they (dis)appear when two neighbors at different positions swap their nearly equal distances to a central atom, while the new distance-based invariant PDD remains continuous.
>
> >The experiment is only done in one database
>
> The Cambridge Structural Database is the world’s largest curated dataset containing 1.17M+ real materials. In any justified approach, experiments are only illustrations for small finite datasets and cannot really replace mathematical proofs because the amount of potential data is often infinite. Even discrete data is so huge (there are more than 4 billion tiny 2x2 grayscale images) that any training on a labeled dataset will always cover only a minuscule proportion of all potential data.
>
> The proposed method to automatically find duplicates among crystals in experimental databases is significant for the growing crisis of fake data, see  https://www.chemistryworld.com/news/800-crystallography-related-papers-appear-to-stem-from-one-paper-mill/4015589.article
>
> >Figure 3 Left: Why can’t we distinguish between these two graphs? The distance matrices of them are different.
>
> Figure 3 shows sets of unlabeled points, not graphs with labelled vertices. The definition of the distance matrix needs labeled points, say indexed by 1,2,...,m,  hence m! permutations to match all distance matrices obtained from m unlabeled points.
>
> >The method is not novel enough. I think other methods such as the radius graph or the combination of radius graph and k-NN graph also suffice to distinguish materials.
>
> Any graph on atomic vertices is discontinuous under perturbations of atoms because of edges that inevitably depend on distance thresholds and detecting neighbors at equal distances in different directions. Hence graph-based methods cannot reliably identify nearly identical crystals.
>
> The key novelty is not in the idea of using k nearest neighbors but in using k neighbors to construct a generically complete invariant that is provably continuous under perturbations. While crystal graphs are restricted to frameworks with rigidly defined bonds, the PDD invariants apply for all periodic (infinite) crystals and even for finite point sets representing rigid (bounded) molecules. Also, we can add atomic types to points but PDD without chemical elements has the ability to compare any crystals even if their chemical compositions are different.
>
> >I recommend using PDD for other material databases to remove variance of your experiments;
>
> The Cambridge Structural Database (CSD) is the most trusted source in materials science because due to the manual curation since the 1960s by the Cambridge Crystallographic Data Centre (CCDC) whose clients include all pharmaceutical giants.
>
> All other datasets are much less reliable, contain simulated (not real) materials, or restrict access by large license fees.
>
> >using PDD as features of other important tasks such as material energy prediction.
>
> This has been done for the averages of columns in PDD matrices, called the Average Minimum Distances, in the paper https://arxiv.org/abs/2108.07233 (published peer-reviewed proceedings).
>
> > limited evaluation
>
> The paper provided the best possible evaluation of the world’s largest and most trusted database of all real crystals (Cambridge Structural Database). The best evaluation is the mathematical proof of all main results: no false negatives, no false positives in general position (always true for noisy data) and continuity under noise.
>
> The major advance is the new continuous approach to classifying periodic structures into infinitely many classes up to isometry, which is the most practical and strongest equivalence relation on solid crystalline materials, while crystallography classified crystals into finitely many classes by discrete tools, for example only 230 crystallographic groups in dimension 3. These group-theoretic classifications are not yet obtained in dimensions higher than 5. All results for the new PDD invariants are proved in any dimension.
>
> The paper proposes a new continuous way to map all crystals in a space with low-dimensional subspaces representing discrete classes.

---

> ### Comment · Reviewer_TtwX · 2022-08-09
> **reply**
>
> I appreciate the authors' efforts to address my concerns. Now I am more convinced by the additional merits of the proposed PDD over graph-based methods for filtering duplicated materials. Hence, I would like to raise my score to 5. Given the weak experiments and relatively narrow research scope to the ML community, I think 5 is sufficient.

---

> > ### Author Response · Authors · 2022-08-09
> > **The significance of the largest possible experiments and the relevance to several Neurips topics**
> >
> > >I appreciate the authors' efforts to address my concerns. Now I am more convinced by the additional merits of the proposed PDD over graph-based methods for filtering duplicated materials. Hence, I would like to raise my score to 5.
> >
> > Thank you for increasing the score.
> >
> > >Given the weak experiments
> >
> > May we ask what experiments were called weak here?
> >
> > Sections 5-6 describe experiments on several databases: the T2 dataset of 5679 simulated crystals reported in the past work [18] Nature, 543:657–664, 2017; and all 660K+ real periodic crystals from the Cambridge Structural Database (CSD).
> >
> > The latter experiment is unique not only by the size of over 200 billion comparisons but is also significant due to ultra fast speed: only two days on the desktop with specifications AMD Ryzen 5 5600X (6-core) @4.60Ghz, 32GB DDR4 RAM @3600 Mhz), see line 249, as expected by the near-linear time complexity in both inputs proved in Theorem 5.1.
> >
> > All found duplicates among real crystals are a real example of identical "needles in haystack" because there were no fast and continuous invariants to detect such duplicates in any large database. If you know another real example of finding such "needles in a haystack" with relatively small resources (in any dataset of hundreds of thousands of hard-to-compare data objects), not necessarily crystal structures, we would be grateful for a reference.
> >
> > >and relatively narrow research scope to the ML community, I think 5 is sufficient.
> >
> > Concerning the research scope at https://neurips.cc/Conferences/2022/CallForPapers, do you agree that the submitted paper is relevant to at least four Neurips topics: Applications, Machine Learning for Sciences (e.g. biology, physics, health sciences, social sciences), Theory, Infrastructure (e.g., datasets, competitions, implementations, libraries)?
> >
> > The submitted paper provides an exemplary approach to solving any real Data Science problem for the following reasons.
> >
> > First, the problem of data classification is rigorously stated via a well-defined equivalence relation: rigid motion or isometry of periodic point sets, which is practically motivated by the rigidity of solid crystalline materials. This definition of an equivalence applies to all (infinitely many) possible data, without any dependence or bias in a labeled finite dataset.
> >
> > Second, the isometry classification problem for periodic structures is solved by explicitly defining the generically complete invariants PDD (Pointwise Distance Distribution), which have clear interpretations via inter-atomic distances, again without relying on any black box optimization, which often finds a local extremum, not a global one.
> >
> > Third, despite their simplicity, the proposed invariants are theoretically proved to be generically complete (injective) in the sense that they always (with 100% certainty) distinguish all non-singular structures.
> >
> > Fourth, the near-linear asymptotic complexity has small hidden constants leading to the lightning fast speed for comparing average structures in nanoseconds on a modest desktop.
> >
> > Finally, the continuously parameterized map of the Crystal Isometry Space is the major advance in materials science after Mendeleev's table of chemical elements because any known and not yet discovered crystal has a unique geographic-style.
> >
> > Hence this "treasure" map visualizes all known classes of materials (for example, putting near-duplicates in small clusters), see first maps in ref [9], and also shows unexplored areas (black holes in this universe of crystals), which might contain future materials with extraordinary properties. Almost all new materials are still designed by trial and error, or by modifying existing ones, hence only slowly exploring the previously unknown map of CRISP.
> >
> > The PDD invariants have already helped discover a new nano-porous crystal in the paper "Analogy Powered by Prediction and Structural Invariants: Computationally Led Discovery of a Mesoporous Hydrogen-Bonded Organic Cage Crystal", J Amer. Chem. Society, 2022, 144, 22, 9893–9901, available at https://pubs.acs.org/doi/pdf/10.1021/jacs.2c02653.
> >
> > In conclusion, the proposed method should become an exemplary approach to any real Data Science challenge: state the classification via a well-defined equivalence relation, then design complete (injective) and fast invariants to distinguish all possible data without relying on inevitably biased finite sets of labeled data, which are always tiny in comparison with all possible data in real-life applications.

---

### Official Review · Reviewer_FXDP · 2022-07-11

**Rating:** 6
**Confidence:** 4
**Soundness:** 3 good
**Presentation:** 3 good
**Contribution:** 3 good

**Summary:**

This paper provides a method based on the Point-wise Distance Distribution PDD(S; k) to describe/characterize periodic crystal arrangements, i.e. atomic/molecular configurations.
This can be used to efficiently identify instances of similar periodic crystal patterns in large datasets.
Essentially, to find a function I on all periodic sets of unlabeled points in Rn such that invariance, completeness, metricity, continuity, computability and inverse design criteria hold.
Proofs are shown.
Various experiments demnostrate the system on atomic crystal configurations from the Cambridge Crystallographic Data Centre (CCDC).

**Questions:**

Is this method relevant to general volumetric AI, where geometric structure is generally not be periodic?

Other approaches such as the Hough transform can be used to identify and classify non-periodic 3D atom-like structures with high accuracy, which invariance to scale, translation, rotation. It seems such a method work well in predicting the periodic structures shown here, also in a manner invariant to scaling. Is this the case?
Chauvin, L., et al. 2022. Registering Image Volumes using 3D SIFT and Discrete SP-Symmetry. arXiv preprint arXiv:2205.15456.




**Limitations:**

Limited to periodic atomic crystal structures.

Alternative methods based on non-periodic atomic 3D image structures may outperform this method.

**Strengths And Weaknesses:**

Strengths: the method apparently works well for indexing crystals, proofs are shown as to the method here, based on distance distributions.

Weakness: predicting periodic crystal structures is a niche domain, it is unclear how this is relevant to general DL.
   Lack of citation to directly relevant research.

---

> ### Author Response · Authors · 2022-08-02
> **The proposed continuous invariants PDD (Pointwise Distance Distributions) also apply to finite or bounded rigid objects in Computer Vision or Graphics**
>
> Thank you for the supportive and objective review.
>
> >This paper provides a method based on the Point-wise Distance Distribution PDD(S; k) to describe/characterize periodic crystal arrangements.
>
> The key result is the *completeness (injectivity)* of PDD for all periodic point sets in general position modulo isometry.
>
> >Predicting periodic crystal structures is a niche domain, it is unclear how this is relevant to DL.
>
> The invariant-based approach goes far ahead of any machine learning by rigorously proving that the proposed PDD invariants have no false negatives (always, for any periodic point sets) and no false positives (for all known periodic crystals in the world's largest database of 1.17M+ real crystals and theoretically proved for all periodic crystals in general position).
>
> >Is this method relevant to general volumetric AI, where geometric structure is generally not be periodic?
>
> You are right that the distance-based invariants PDD are well-defined for any finite sets of points and (similarly to the periodic case) continuously change under perturbations in the Earth Mover's Distance. Hence the PDD invariants can be efficiently used for comparing any finite or founded rigid objects (in Computer Vision or Graphics) given by their feature points.
>
> The periodic structures are the most important practical case with more than 660K+ real periodic crystals in the Cambridge Structural Database. More general crystals (quasiperiodic, disordered, etc) have no formal definitions and standard representations. The periodic case is already huge enough, and should be resolved first.
>
> >Other approaches such as the Hough transform can be used to identify and classify non-periodic 3D atom-like structures with high accuracy, which invariance to scale, translation, rotation.
>
> Our work focuses on periodic point sets, while the Hough transform requires a finite set of pixel-based points in a bounded image. High accuracy can be achieved on any finite dataset by optimizing and over-fitting parameters long enough. However, the PDD invariants have stronger guarantees of generic completeness (provably distinguish all non-singular crystals) without any known counterexamples among real periodic crystals.
>
> >a method work well in predicting the periodic structures shown here, also in a manner invariant to scaling. Is this the case?
>
> You are right that a uniform scaling of a periodic structure by a scalar factor s multiplies all elements of the PDD invariant by s, hence normalising PDD gives an invariant up to isometry and uniform scaling.
>
> >Chauvin, L., et al. 2022. Registering Image Volumes using 3D SIFT and Discrete SP-Symmetry. arXiv preprint arXiv:2205.15456.
>
> This paper appeared on arxiv after the Neurips deadline and considers invariance only under “inversion of spatial axes” (see their abstract), not under infinitely many rotations.
>
> >Limited to periodic atomic crystal structures.
>
> This is not a limitation because periodic crystals are the most general case of real crystalline materials. A quasi-periodic crystal is a sliced projection of a high-dimensional periodic structure. Since the PDD invariants are defined in any dimension, they can be extended to more general materials. Theorem 13 in ref [20] about the asymptotic behavior of distances to neighbors (rows in the PDD matrix) is already proved for a more general case including non-periodic sets.
>
> >Alternative methods based on non-periodic atomic 3D image structures may outperform this method.
>
> We would appreciate a reference for comparable alternatives. However, atomic 3D image structures are restricted to bounded domains while PDD invariants are generically complete due to Theorem 4.4 in the much harder case of infinite periodic structures.
>
> It seems unrealistic to outperform PDD because they distinguished all real periodic crystals in the world’s largest database curated by the Cambridge Crystallographic Data Centre, which considers all our found duplicates as suspicious and actually asked five journals to investigate the data integrity of the underlying publications.
>
> >moderate-to-high impact paper
>
> Many colleagues are highly positive about the impact because the PDD invariants justified that all real periodic crystals live in a common Crystal Isometry Space expending Mendeleev’s table of elements to all periodic crystals, which is a major advance in Data Science.

---

> > ### Comment · Reviewer_FXDP · 2022-08-06
> > **Incorrect dismissal of related work**
> >
> > In my reviewer, I mentioned that the authors consider the 3D SIFT method, which been used for over a decade [a,b,c] to robustly align and identify particle arrangements in 3D images. It is fully invariant to 3D rotation, translation and scaling of the image, and would possibly work quite well in the authors specific context here, as mentioned, for this very reason.
> >
> > The authors unfortunately choose to dismiss this literaure, incorrectly saying it is not invariant to rotation. It is true reference [a] recently extends invariance to sign-parity transforms, however it is invariant to 3D similarity transforms (including 3D rotation, translation and scaling) like other previous work [b,c]
> >
> > [a] Chauvin, L., et al. 2022. Registering Image Volumes using 3D SIFT and Discrete SP-Symmetry. arXiv preprint arXiv:2205.15456.
> >
> > [b] B. Rister et al. "Volumetric image registration from invariant keypoints." IEEE Transactions on Image Processing 26.10 (2017): 4900-4910.
> >
> > [c] M. Toews et al. "Efficient and robust model-to-image alignment using 3D scale-invariant features." Medical image analysis 17.3 (2013): 271-282.

---

> > > ### Author Response · Authors · 2022-08-06
> > > **The case infinite periodic sets of points substantially differs from the case of finite point sets**
> > >
> > > >Incorrect dismissal of related work
> > >
> > > We are genuinely surprised to read such a strong statement because we did not use any strong words such as "dismiss".
> > >
> > > When the latest paper [a] arXiv:2205.15456 was cited in the original comment, we replied that it was first submitted to the arxiv on 30th May 2022 (after the Neurips deadline). Do you agree that the original submission cannot be expected to discuss the work appearing after or shortly before the deadline?
> > >
> > > >In my reviewer, I mentioned that the authors consider the 3D SIFT method, which been used for over a decade [a,b,c] to identify repeated particle arrangements in 3D images.
> > >
> > > More importantly, we did read the paper and replied that this work is restricted to non-periodic inputs. Here is the key quote from page 1 of paper [a], also in the abstract: "We refer to this as a sign inversion and parity transform (SP-transform), which applies generally to arbitrary scalar image fields I : R^3->R^1 mapping 3D coordinates (x; y; z) in R^3 to a scalar image intensity I(x; y; z) in R^1, and we seek SP-invariance". Do you agree that this paper studies scalar 3D images, not infinite periodic sets of points?
> > >
> > > >It is fully invariant to 3D rotation, translation and scaling of the image, and would possible work quite well in the authors specific context here, as mentioned, for this very reason.
> > >
> > > Figure 4 (left) in the original submission shows that there is no easy way to reduce a periodic point set to its finite subse without losing invariance up to isometry because all standard approaches by taking a box or a ball with a fixed cut-off radius produce non-isometric subsets of the same periodic set.
> > >
> > > Paper [b] "Volumetric image registration from invariant keypoints" follows the same cut-off aporoach with the radius σ on page 7: "To compute the keypoint descriptor, also known as the feature vector, we first take a spherical image window centered at the keypoint, of radius 2σ, where σ is a constant multiple of the keypoint scale from equation 2. To achieve rotation invariance, the image is rotated by the inverse of the keypoint orientation from section III-B. The spherical window is then divided into a 4 × 4 × 4 array of cubical sub-regions, as seen in figure 5."
> > >
> > > Slight perturbations can easily push points closer or further away from the spherical image window of a fixed size. Hence continuity (needed for most opmitizations in machine learning) of the resulting descriptor under perturbations of points requires extra justifications. Continuity requirement in Problem 1.1(d) is the key novelty in the submitted paper.
> > >
> > > >It is true reference [a] recently extends invariance to sign-parity transforms, however it is invariant to 3D similarity transforms (including 3D rotation, translation and scaling).
> > >
> > > You are right that the invariance under 3D rotation followed from the previous work. However, this approach was developed for finite sets of feature points, not for infinite periodic sets.
> > >
> > > Do you agree, that similar to the other mentioned papers [a,b], paper [c] "Efficient and robust model-to-image alignment using 3D scale-invariant features" studies finite sets of feature points obtained from local spherical images?
> > >
> > > Here is the quote from section 2.1 on page 6: "Invariant feature extraction begins by identifying a set of location/scale pairs {(Xi, σi)} in an image. This is done by detecting spherical image regions centered on location Xi with radius proportional to scale σi that locally maximize a function f(X, σ) of image saliency."
> > >
> > > We are happy to include the proposed references at the beginning of section 2.1 in the submitted paper, where the discussion of the finite case reviews the simpler and faster descriptor (total pairwise distribition of pairwise distances), which is not only invariant under isometry (compositions of translations, rotations, reflections) in any dimension and can be made invariant under uniform scaling by normalization, also continuous under perturbations, but is also generically complete (injective) by distinguishing all non-isometric finite sets in general position.
> > >
> > > The counter-examples to completeness of the above invariant in Figure 3 were a key motivation to develop a stronger Pointwise Distance Distribution (PDD), which is generically complete in the much harder periodic case.
> > >
> > > If anything remains unclear, we would be happy to provide more details.

---

> > > > ### Comment · Reviewer_FXDP · 2022-08-08
> > > > **followup**
> > > >
> > > > * Thanks for your response. I understand that the proposed approach is specialized towards identifying periodic crystal structures from periodic sets of points, and not scalar images as in the 3D SIFT approach. I view this work as a specialized analysis of 3D periodic point geometry, application to periodic atomic crystals, and within this context it is interesting.
> > > >
> > > > > You are right that the invariance under 3D rotation followed from the previous work. However, this approach was developed for finite sets of feature points, scalar images, not for infinite periodic sets.
> > > >
> > > > * It is worth mentioning the SIFT method has been used to identify 3D point set configurations for a decade in a manner fully invariant to 3D rotations. In the case of 3D atomic point set arrangements (periodic or non-periodic), a single preprocessing step is required: convert the point set (either periodic or non-periodic) to a scalar electron density image (so-called "dft" image approximating the Schrodinger equation). Afterwards, standard SIFT descriptor voting/clustering can be used to identify solutions. That being said, I do not see a reference using 3D SIFT for the specific task of identifying periodic atomic crystal arrangments.
> > > >
> > > > >  Slight perturbations can easily push points closer or further away from the spherical image window of a fixed size. Hence continuity (needed for most opmitizations in machine learning) of the resulting descriptor under perturbations of points requires extra justifications. Continuity requirement in Problem 1.1(d) is the key novelty in the submitted paper.
> > > >
> > > > * While the authors solution to continuity here appears novel when working solely from periodic point set data, it seems there are other more general solutions available to the same specific task. Continuity, fixed window sizes are not issues in the SIFT algorithm. Continuity is ensured by Gaussian constraints, and the size of of spherical windows is not fixed, it is determined automatically for each keypoint as maximizing the local image energy, and the resulting point descriptors are highly stable to perturbations.

---

> > > > > ### Author Response · Authors · 2022-08-08
> > > > > **If any past results are comparable to Theorems 4.3, 4.4, 5.1, then references to exact statements would be helpful**
> > > > >
> > > > > Thank you for your detailed reply.
> > > > >
> > > > > >I view this work as a specialized analysis of 3D periodic point geometry, application to periodic atomic crystals, and within this context it is interesting.
> > > > >
> > > > > Thank you for highlighting the interest. The new methods are developed for all dimensions, not only 3D. Figures 1, 2, and 4 (left) highlight that there is no easy reduction of the periodic case to the finite case because a basis for an underlying periodic structure can be chosen in infinitely many different ways. Theorem 15 in past work [20] formally proves that all attempts to define a unique reduced basis are discontinuous under perturbations. Hence the continuity of any isometry invariants in the finite case cannot guarantee the continuity in the periodic case.
> > > > >
> > > > > The crisis of fake data in crystallography and materials science (https://www.chemistryworld.com/news/800-crystallography-related-papers-appear-to-stem-from-one-paper-mill/4015589.article) is growing because there were no reliable (fast and continuous) way to compare periodic structures in large experimental databases.
> > > > >
> > > > > The key practical achievement (section 6, lines 295-315) consists of the 220B+ pairwise comparisons of all 660K+ periodic crystals in the world's largest database of real materials, completed only over two days on a modest desktop, while the traditional RMSD tool (not producing a continuous metric) requires the estimated 38 days to compare one crystal with all others, hence 38 x 660 K days / 2 > 343 thousand years for all comparisons on the same machine.
> > > > >
> > > > > Since this experiment revealed only suspicious duplicates (now investigated by five journals), the completeness of PDD invariants currently holds for all periodic crystals without any known counter-examples. Moreover, the past work [3] provided slower but provably complete isometry invariants for all periodic point sets, which can be used to distinguish any potential singular structures if they emerge in the future.
> > > > >
> > > > > These results led to the major discovery in data science that all real periodic crystals have unique (geographic-like) locations in a common Crystal Isometry Space (CRISP) continuously extending Mendeleev's table of elements. Hence a continuous map of CRISP provides an efficient guide to search for new materials instead of blind sampling.
> > > > >
> > > > > >In the case of 3D atomic point set arrangements (periodic or non-periodic), a single preprocessing step is required: convert the point set (either periodic or non-periodic) to a scalar electron density image (so-called "dft" image approximating the Schrodinger equation).
> > > > >
> > > > > Does this conversion require extra parameters? Is it correct that a discretized image of a density function might be more complicated than an initial finite set of points? If we need to recognize point sets up to isometry or rigid motion, which is the most practical equivalence for solid crystals, the simpler and faster way to solve Problem 1.1 (lines 55-65 in the submitted paper) can use only distances to neighbors, which are computationally easy (due to the well-studied nearest neighbor problem) and stable under perturbations. Does the SIFT solve a different problem?
> > > > >
> > > > > >Afterwards, standard SIFT descriptor voting/clustering can be used to identify solutions.
> > > > >
> > > > > Does this voting/clustering require more parameters and how are they chosen?
> > > > >
> > > > > >That being said, I do not see a reference using 3D SIFT for the specific task of identifying periodic atomic crystal arrangments.
> > > > >
> > > > > It is possible that 3D SIFT tries to solve another problem, so we would be grateful for a reference to the problem statement.
> > > > >
> > > > > >Continuity, fixed window sizes are not issues in the SIFT algorithm. Continuity is ensured by Gaussian constraints,
> > > > >
> > > > > Is it correct that a Gaussian constraint needs at least one extra parameter and how is it chosen?
> > > > >
> > > > > > and the size of of spherical windows is not fixed, it is determined automatically for each keypoint as maximizing the local image energy
> > > > >
> > > > > Does this automatic selection guarantee no loss of information at least in general position so that almost different images (or sets of feature points) are always distinguished in all non-singular cases?
> > > > >
> > > > > >and the resulting point descriptors are highly stable to perturbations.
> > > > >
> > > > > Would it be possible to give exact references to specific statements and proofs for the SIFT analogs of the following new results in the submitted paper?
> > > > >
> > > > > Theorem 4.3 (lines 229-230): Lipschitz continuity of the proposed invariant PDD (Pointwise Distance Distribution).
> > > > >
> > > > > Theorem 4.4 (lines 242-245): generic completeness of PDD in the sense that all non-singular periodic point sets are distinguished by PDD with a specific upper bound for a required number k of neighbors.
> > > > >
> > > > > Theorem 5.1 (lines 256-258): near-linear asymptotic complexity in both inputs (k and number m of points in a finite set or in a motif of a periodic set) for a fixed dimension n.
> > > > >
> > > > > We would be happy to add references to all results that are comparable with the new theorems above. Thank you.

---

### Meta-Review · Area_Chair_MaG7 · 2022-09-13

**Recommendation:** Accept
**Confidence:** Less certain

**Metareview:**

While the paper is not rated too high by the reviewers, and overall the endorsement for it is a bit less strong that what we would have liked to see, it seems that the paper might have results that are of interest to those working on materials science applications of ML.

One common complaint the authors had was the limited experimentation, but given the authors' response regarding the validation (which in particular goes through an important dataset), the concern was overcome. However, to improve the impact of the paper, and to offer stronger motivation for followup works, it would be a valuable addition to the final version of the paper to have more experiments, especially those that more elaborately outline suitable ablation studies, to illustrate the theory in action, and importantly, to add in some comparisons against some of the generic methods (which will be understandably, not respecting all desired invariances, still, it is good to see how much of a big deal that is, notwithstanding the comments on Platon et al).

One minor but potentially useful technical point to note is that what the authors call "continuity" (Theorem 4.3) is a type of Lipschitz continuity, which can be further exploited for algorithm design and analysis.



**Award:**

No

---

### Decision · Program_Chairs · 2022-09-14

Accept